# Long-Term Compost Amendment Changes Interactions and Specialization in the Soil Bacterial Community, Increasing the Presence of Beneficial N-Cycling Genes in the Soil

Jessica Cuartero [1,*], Onurcan Özbolat [2], Virginia Sánchez-Navarro [3], Julia Weiss [2], Raúl Zornoza [3], José Antonio Pascual [1], Juana-María Vivo [4] and Margarita Ros [1]

1   Centre of Edaphology and Applied Biology of the Segura (CEBAS-CSIC), University Campus of Espinardo, 30100 Murcia, Spain; jpascual@cebas.csic.es (J.A.P.); margaros@cebas.csic.es (M.R.)
2   Institute of Plant Biotechnology, Polytechnic University of Cartagena, Plaza del Hospital s/n, 30202 Cartagena, Spain; onurcan.ozbolat@edu.upct.es (O.Ö.); Julia.Weiss@upct.es (J.W.)
3   Department of Agricultural Science, Polytechnic University of Cartagena, Paseo Alfonso XIII 48, 30203 Cartagena, Spain; virginia.sanchez@upct.es (V.S.-N.); raul.zornoza@upct.es (R.Z.)
4   Department of Statistics and Operations Research, University of Murcia, CMN and IMIB-Arrixaca, 30100 Murcia, Spain; jmvivomo@um.es
*   Correspondence: jcuartero@cebas.csic.es

**Abstract:** Significant differences in the microbial community and diversity in soil have been observed due to organic farming, but little research has been performed for exploring microbial functionality and the co-occurrence of patterns among microbial taxa. In this work, we study soil 16S rDNA amplicons from two long-term organic farming systems (Org_C and Org_M) and a conventional system (Conv) to decipher the differences in microbial interaction and network organization and to predict functional genes (principally related to the N cycle). In general, the network organizations were different in all cropping systems due to agricultural management. Org_C showed the highest negative interactions and modularity and the most altered bacterial niches and interactions, which led to an increase in generalist species that stabilize the bacterial community and improve the response of the soil to adverse conditions. These changes altered the predicted functionality of the bacterial community; Org_C showed higher referred numbers of nitrogen fixation genes, a decrease in the $N_2O$ emission genes and could favor the uptake of environmental $CO_2$. Thus, long-term compost amendment application has significant benefits for the farmer and the environment, since prolonged application can reduce the use of fertilizers and pesticides and could create a more stable soil, which could resist the effects of climate change.

**Keywords:** co-occurrence; conventional farming; manure; organic farming; PICRUSt; soil bacteria

## 1. Introduction

In recent decades, the concept of sustainable agriculture has been gaining ground. Sustainable agriculture is an integrated system in which plant production practices have a site-specific application that has long-term benefits [1]. Sustainable agriculture not only satisfies human food and fiber needs, but also enhances environmental and natural resource quality [2]. Crop rotation, intercropping and organic fertilizers are common sustainable practices across a diverse array of agroecosystems. These practices can break cycles of disease and pests, improve soil fertility, suppress weeds and improve food and nutritional security [3,4].

Within sustainable agriculture, organic farming is perceived to be more environmentally friendly than conventional farming while producing higher quality crops and similar yields [5,6]. Soil organic matter content plays an important role in soil fertility, maintaining soil functions and reducing erosion. However, its build-up is a very slow process, whereas

its decline is relatively fast [7]. Soil organic matter usually depends on the input of organic material, and it contributes to the improvement of physicochemical, chemical and biological properties of soils. Compost is one of the best alternatives for incorporating organic matter and nutrients into the soil; it is considered a cheap organic amendment that is agronomically advantageous and environmentally safe, and which stimulates soil microbial activity and crop growth [5,8].

Soil microorganisms maintain soil health and are crucial for crop production in agricultural systems. They play an essential role in the soil structure as well as in decomposing organic matter, degrading contaminants, suppressing soil-borne diseases [9] and fertilizing the soil [10,11]. The complex interaction among different microbial species through the flow of energy, matter and information forms large, complex ecological networks; it is essential to understand the underlying mechanisms in order to fully understand the soil microbiota [12]. Determining microbial network structures and their relationships to environmental changes and metabolic processes in microbial communities is a significant challenge in agricultural soils [13].

Microbial metabolism involves a large set of functional genes and biochemical pathways that power biogeochemical cycling in soil. Nitrogen has been identified as the main yield-limiting nutrient for organic cropping systems [14], so in-depth knowledge of its transformation and fixation in the soil could help to understand which soil microorganisms are best adapted to the soil for a specific crop [15,16].

A good approach for studying the potential functionality of microbial communities has recently been defined. This approach consists of predicting functionality from 16s rRNA gene sequence data based on Phylogenetic Investigation of Communities by Reconstruction of Unobserved States (PICRUSt), which uses an extended ancestral-state construction as a predictor [17]. This procedure is followed by predicting the functional genes associated with nitrogen metabolism based on the Kyoto Encyclopedia of Genes and Genomes database (KEGG). This method has gained popularity in recent years in ecology [18,19] and agriculture [20,21], but the limitations of a metagenomic technique must be taken into account. These limitations are mainly due to the short fragment length and potential inaccuracies in the prediction process, so we should keep these factors in mind as we consider potential functionality.

To analyze the interactions among different microbial communities, a network-based bioinformatics approach has been used. This approach is based on high-throughput metagenomics sequencing data, where the network is a representation of various biological interactions, e.g., predation, competition and mutualism in soil in which species (nodes) are connected by pairwise interactions (edges) [22,23].

Future agriculture needs to focus on measures that improve soil biological functions for appropriate soil health management [24]. Functional traits and phylogenetic network analyses are valuable ecological markers for understanding microbial community assembly, and they help elucidate how natural communities and their functions respond to environmental and soil management changes [25–27].

We hypothesized that bacterial communities become stable over time under different cropping systems and that bacterial communities under long-term organic farming systems will be more diverse and will show a greater capacity to adapt to external agents. The objectives of this work were: (i) to explore the effect of farming system on network interactions among different phylogenetic groups by phylogenetic molecular ecological networks (pMENs); (ii) to study the predicted functions of an organic farming system compared to a conventional system, principally the functional genes associated with nitrogen cycle; and (iii) evaluate the relationships among bacterial modules and physicochemical soil properties.

## 2. Materials and Methods

### 2.1. Experimental Design and Sampling

The soil for the study was Haplic Calcisol (loamic, hypercalcic) IUSS [28], located in Campo de Cartagena in Murcia, southeastern Spain. Since the early 1990s, fifteen plots have been used for vegetable cultivation with three different cropping systems (five plots for each): (1) a conventional system (Conv); (2) an organic system with a yearly addition of compost and compost tea (Org_C); and (3) an organic system with a yearly addition of sheep manure (Org_M). More information can be found in Table S1. The characteristics of the sheep manure, compost and compost tea were previously described in Cuartero et al. [29]. The sampling was carried out in February 2018 after the harvest of a leaf cabbage crop (Brassica olearacea var. sabellica) grown during the winter season. One composite sample derived from 10 subsamples (0–10 cm depth) was collected with an auger from each plot.

Samples were taken to the lab immediately and separated into two aliquots. The soil for biological analysis was sieved at <2 mm and stored at −20 °C, and the soil for physicochemical and chemical analyses was sieved at <2 mm and kept at 4 °C. Soil properties were measured according to Cuartero et al. [29].

### 2.2. Soil Properties, DNA Extraction, Sequencing, Data Processing and Function Prediction

Total organic carbon (TOC), pH, electrical conductivity (EC), total nitrogen (TN), $NH4^+$, available P, Mg, Na and Ca were measured according to Cuartero et al. [29]. Soil DNA extraction, bacterial community analysis (by next generation sequencing of bacterial 16S hypervariable regions) and data processing were performed according to Cuartero et al. [29] using the Greengenes database. The sequences are available at the European Nucleotide Archive (ENA) with the study accession code PRJEB38121.

The metagenomes were predicted from the OTU table using Phylogenetic Investigation of Communities by Reconstruction of Unobserved States (PICRUSt) [17]. The OTUs were normalized by dividing each OTU by the predicted 16S copy number abundance, and the functional genes were identified based on Kyoto Encyclopedia of Genes and Genomes (KEGG) pathways [30].

### 2.3. Construction and Analysis of the Microbial Network

Network analyses were performed to discern co-occurrence patterns of the soil microorganisms by constructing phylogenetic molecular ecological networks (pMENs) through the open-access molecular ecological network analysis pipeline (http://ieg4.rccc.ou.edu/mena/, accessed on 17 January 2022) [12,13,31]. Network construction was based on relative abundance in the soil samples for each cropping system. The relative abundances of OTUs were transformed into log matrixes, and a Pearson correlation matrix was estimated. A reliable similarity threshold (St) for the correlation matrix based on the χ2-test with Poisson distribution was automatically identified according to a Random Matrix Theory (RMT)-based approach prior to the network construction. The adjacency matrix was derived only from OTUs with similarity values above an optimal St. These OTUs were represented as nodes, and their pairwise interactions were represented as edges. For each of the resulting pMENs, 100 randomly rewired networks were generated, keeping the network size and number of links. Welch's t-test followed by a Games—Howell post-hoc test were performed to compare topological network properties under different cultivation systems from the standard deviations reported from their respective random networks. Network modules were detected using the fast greedy modularity optimization method. Eigengene network analysis was carried out by performing singular value decomposition (SVD) in order to summarize each module with a single representative abundance profile, known as the module eigengene. Small modules with fewer than five nodes were not used. Moreover, hierarchical cluster trees and heatmaps were derived to display the module eigengenes of each module, higher-order organization and correlations between modules. Identification of key module members (MMs) was based on threshold values of $Z_i$ (within-module connectivity) of 2.5 and $P_i$ (among-module connectivity) of 0.62 accord-

ing to Guimera and Nunes Amaral [32] and Olesen et al. [33]. OTU roles can be thereby categorized into peripherals ($Z_i < 2.5$; $P_i < 0.62$), connectors ($Z_i < 2.5$; $P_i \geq 0.62$), module hubs ($Z_i \geq 2.5$; $P_i < 0.62$) and network hubs ($Z_i \geq 2.5$; $P_i \geq 0.62$). Furthermore, Mantel tests were performed to detect relationships under the three cropping systems between network connectivity, soil properties and metabolic functionalities. OTU significance was calculated previously. The networks were visualized using Cytoscape software version 3.5.1 [34] and the ggplot2 package [35].

### 2.4. Statistical Analysis

To evaluate the effect of cultivation systems on OTU variations, permutational multivariate analysis of variance (PERMANOVA) was conducted using the 'betadisper' and 'adonis' functions with 999 permutations from the vegan package version 2.5-7 [36]. Furthermore, functional profile assignments from PICRUSt were also tested with R version 4.0 [37]. Normality and homogeneity of variance assumptions were assayed by Shapiro—Wilk and Bartlett's tests. Mean comparisons were performed with one-way analysis of variance (ANOVA) followed by post-hoc tests, Tukey's honestly significant difference (HSD) for all-pair comparisons and Dunnett's comparisons for the control system. In cases in which homoscedasticity was not met, Welch's t-test was performed using the 'pairwise.*t*.test' function with Bonferroni—Holm correction for multiple comparisons. The robustness of the estimations was checked by the bootstrapping approach using 1000 replicates. When data did not fit a normal distribution, non-parametric Kruskal—Wallis tests were performed, and if the assayed data were significant, a multiple comparison Z-values test was performed using the 'dunnTest' function with Benjamini—Hochberg corrections in the FSA package version 0.8.30 [38].

## 3. Results

### 3.1. Network Analysis

After data preprocessing, 539 OTUs remained in both the Conv and Org_C data sets for network construction, and 439 OTUs remained in the Org_M data set. Optimal similarity thresholds for the correlation matrixes obtained were identical (0.97) for the three soil microbial communities. Applying such a cut-off, two networks of similar size, nodes and links were constructed for Conv and Org_C, and another one of a smaller size was constructed for Org_M (Table 1). In addition, network connectivity distribution curves fitted well with the power law model ($R^2$ varied from 0.75 from 0.83).

**Table 1.** Topological properties of the empirical pMENs of microbial communities of the three cropping systems and their associated random pMENs.

| Treatment | Empirical Networks | | | | | | | | | | Random Networks | | |
| | No. of Original OTUs | Similarity Threshold St | Network Size | R Square of Power-Law | Avg Connectivity | Node | Edge | Average Path Distance (GD) | Avg Clustering Coefficient | Modularity (No. of Modules) | Avg Path Distance ±SD | Avg Clustering Coefficient ± SD | Avg Modularity ± SD |
| --- | --- | --- | --- | --- | --- | --- | --- | --- | --- | --- | --- | --- | --- |
| Conv | 539 | 0.97 | 453 | 0.83 | 6.04 | 404 | 1220 | 6.821 b | 0.368 b | 0.646 (32) c | 3.430 ± 0.031 | 0.037 ± 0.004 | 0.369 ± 0.005 |
| Org_C | 539 | 0.97 | 452 | 0.80 | 6.80 | 400 | 1360 | 6.896 b | 0.395 a | 0.698 (36) b | 3.300 ± 0.031 | 0.044 ± 0.005 | 0.337 ± 0.005 |
| Org_M | 439 | 0.97 | 396 | 0.75 | 4.08 | 357 | 729 | 8.262 a | 0.355 c | 0.824 (29) a | 4.176 ± 0.038 | 0.014 ± 0.04 | 0.497 ± 0.008 |

Values followed by different letters represent significant differences between cropping systems by Games—Howell's post-hoc test: Conv, conventional system; Org_C, organic cultivation with sheep manure compost and compost tea; Org_M, organic cultivation with sheep manure.

Data revealed modularity values higher than 0.4. The highest connectivity and clustering coefficient was found for the Org_C network, followed by the Conv and Org_M networks. The average path distance and modularity were lower for Conv, followed by the Org_C and Org_M networks (Table 1). In addition, the average path distance and modularity of the networks were larger than their respective random networks (Table 1).

Overall network indices for the identified pMENs under the distinct cultivation systems reported significant differences. We found that the Org_C network was composed of 400 nodes (OTUs) linked by 1360 edges (789 positive edges and 571 negative edges); Conv

by 404 nodes and 1220 edges (875 positive edges and 345 negative edges); and Org_M by 357 nodes and 729 edges (453 positive edges and 276 negative edges). The Org_C network showed 15 major modules (modules with more than 5 nodes), followed by Conv with 13 modules and Org_M with 12 modules (Figure 1).

All of the nodes included in the major modules had significant ($p < 0.05$) module memberships (MMs), as shown by module eigengene analysis. A total of 1015 significant MMs were observed, of which 703 were shared between the three cropping systems, accounting for 59% of the Org_C network, 65% of the Org_M network and 57% of the Conv network. In general, Actinobacteria and Proteobacteria were the dominant phyla in the three networks, but Firmicutes, Bacteriodetes, Acidobacteria and Verrucomicrobia were widely distributed as well (Table S2).

Moreover, eigengene analysis, based on the clustering dendrogram and heat map, revealed differences in higher-order organization between the networks (Figure 1). Module eigengenes explained 57–96%, 58–92% and 59–89% of the variations of relative abundance across different samples in the Org_C, Conv and Org_M networks, respectively (Figure 1). Eigengenes from modules showed significantly higher correlations among modules in the Org_C (OC5-OC7; OC3-OC15; OC14-OC13; OC4-OC12 and OC12-OC13) and Conv (C1-C13 and C9-C12) networks, whereas Org_M did not show significant correlations among modules (Figure 1).

The eight OTUs with the highest abundance (436; 605; 110; 003; 324; 438; 486 and 174) and their nearest neighbors were selected for study of the connection variations in a subnetwork. The Org_C subnetwork had the highest number of connections (50 nodes and 130 edges), followed by Org_M (47 nodes and 111 edges) and Conv (43 nodes and 91 edges) (Figure 2). The most abundant OTUs showed more direct connections and interactions in Org_C, with 46 nodes and 46 edges (28 positives and 18 negatives), than in Org_M, with 40 nodes and 40 edges (37 positive and 3 negative), or Conv, with 35 nodes and 35 edges (12 positives and 23 negatives) (Table S3).

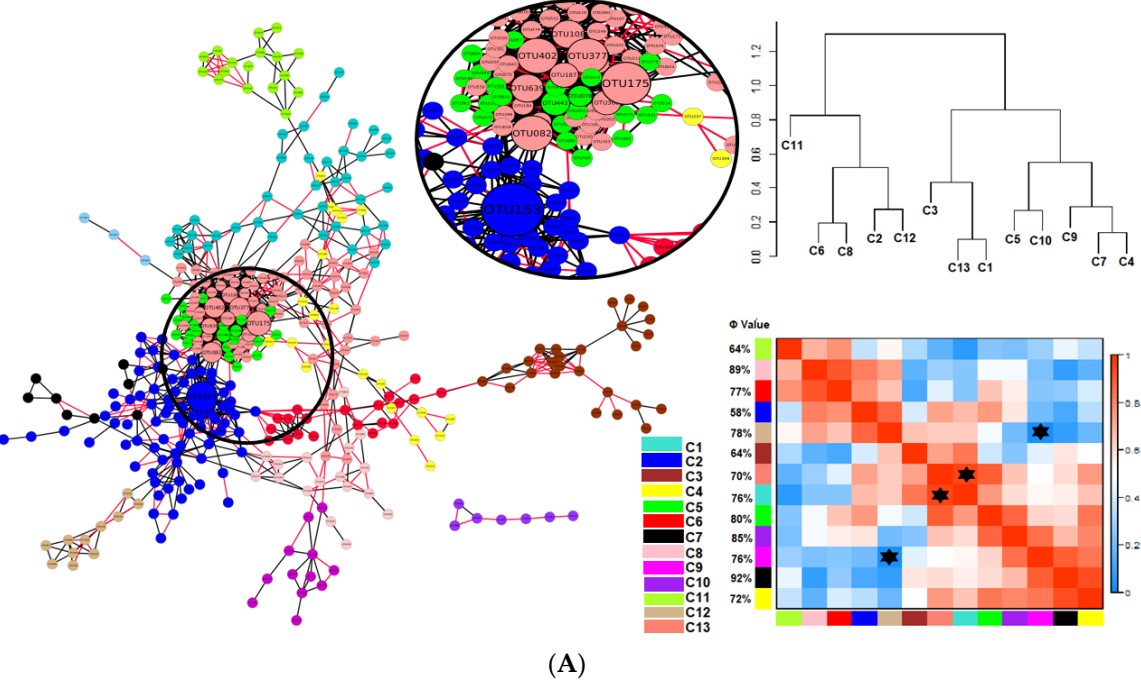

(**A**)

**Figure 1.** *Cont.*

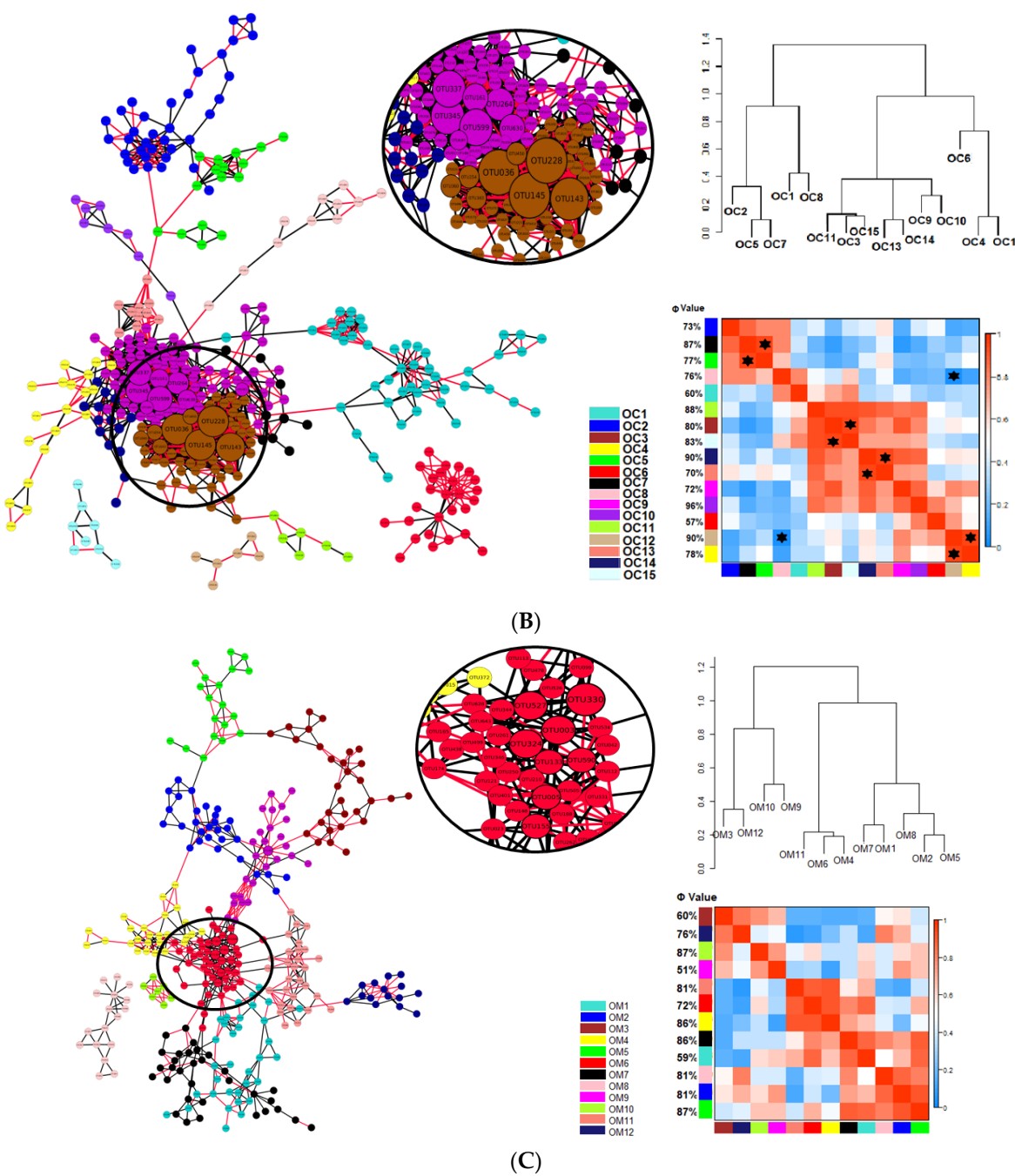

**Figure 1.** Networks of (**A**) the conventional cropping system (Conv) and both organic cropping systems, (**B**) Org_C and (**C**) Org_M, based on OTU profiles. On the top right a hierarchical clustering is shown based on Pearson correlations among module eigengenes. Below the clustering, a heatmap shows the coefficient values. The color red means higher correlation, while blue signifies lower correlation. Modules smaller than five nodes were excluded from eigengene analysis and are not displayed. Modules larger than five nodes are labelled with different colors in the network. The size of the circle is directly proportional to the number of edges in the module, and a large circle implies a higher number of connections in the node. Red and black lines represent negative and positive edges, respectively. A black star on the heatmap (★) indicates a significant (*p* < 0.05) correlation among modules: Conv, conventional cropping system; Org_C, organic cropping system with compost and compost tea; Org_M, organic cropping system with manure.

### 3.2. The Generalist Presence in Networks

A Zi-Pi plot was constructed to illustrate the topological roles of individual network nodes (Figure 3). The majority of the OTUs (>98%) observed in the three pMENs were categorized as peripherals (representing specialist nodes from an ecological perspective) with most of their edges inside their own modules (71.6% for Conv, 75.3% for Org_C and 82.3% for Org_M) (Figure 3). Few module hubs (generalists) were present (1% for Conv, 1.5% for Org_C and 1.7% for Org_M). In addition, a total of two nodes (0.5%) were connectors (generalists) for the Conv and Org_C networks, whereas none were identified for the Org_M network. No network hubs (supergeneralists) were observed in any of the three networks.

Six module hubs belonging to Actinobacteria (*Agromyces* and Solirubrobacterales), Proteobacteria (*Arenimonas*, *Ramlibacter* and *Geobacter*), Planctomycetes, Chloroflexi and NKB19 were observed in the Org_C network. Five module hubs belonging to Actinobacteria, Protobacteria, Firmicutes (*Bacillus*) and Verrucomicrobia were found in the Org_M network. In the Conv network, four module hubs were found belonging to Actinobacteria (*Nocardioides* and *Rubrobacter*) and Proteobacteria (*Methylotenera* and *Sphingopyxis*) (Table S4). Connectors in the Org_C network (OTU461 and OTU508) belonged to Proteobacteria, and those in the Conv network (OTU085 and OTU442) belonged to Actinobacteria and Proteobacteria (Table S4). Interestingly, some of the nodes inverted their topological function, serving as a generalist in one network and a specialist in another (Table S4).

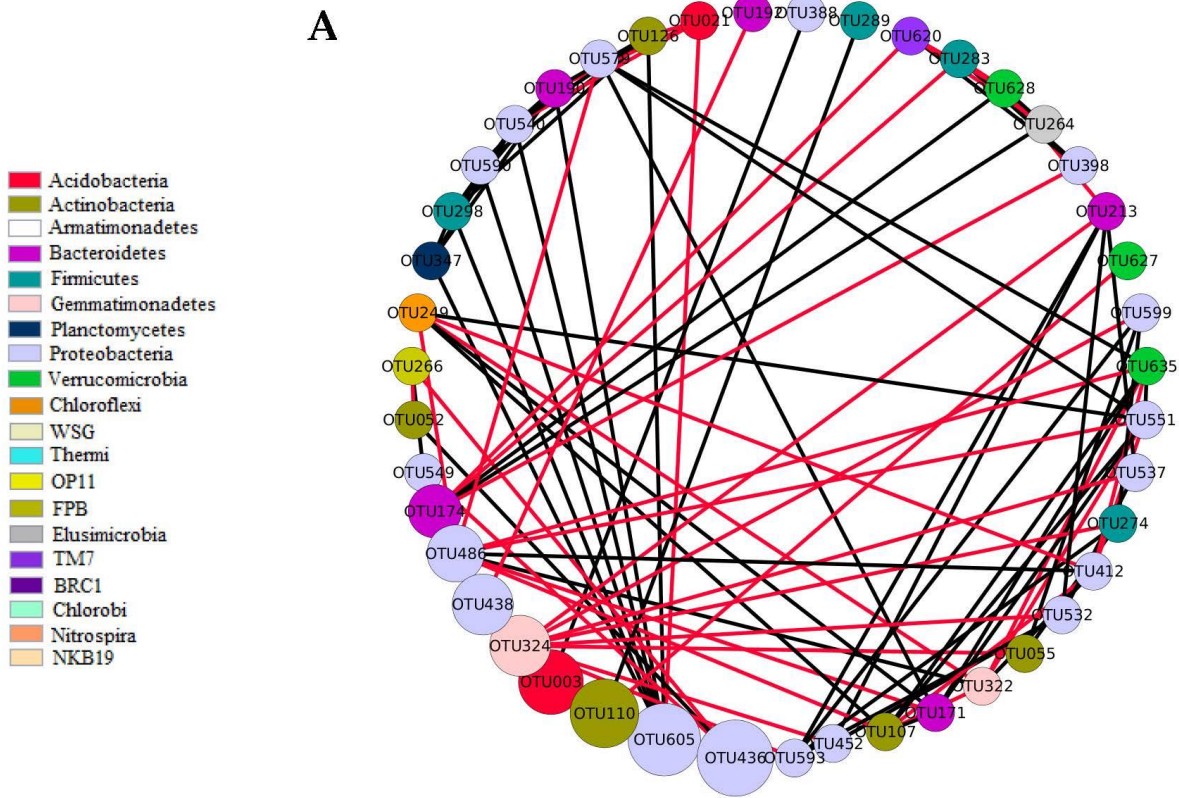

**Figure 2.** *Cont.*

**B**

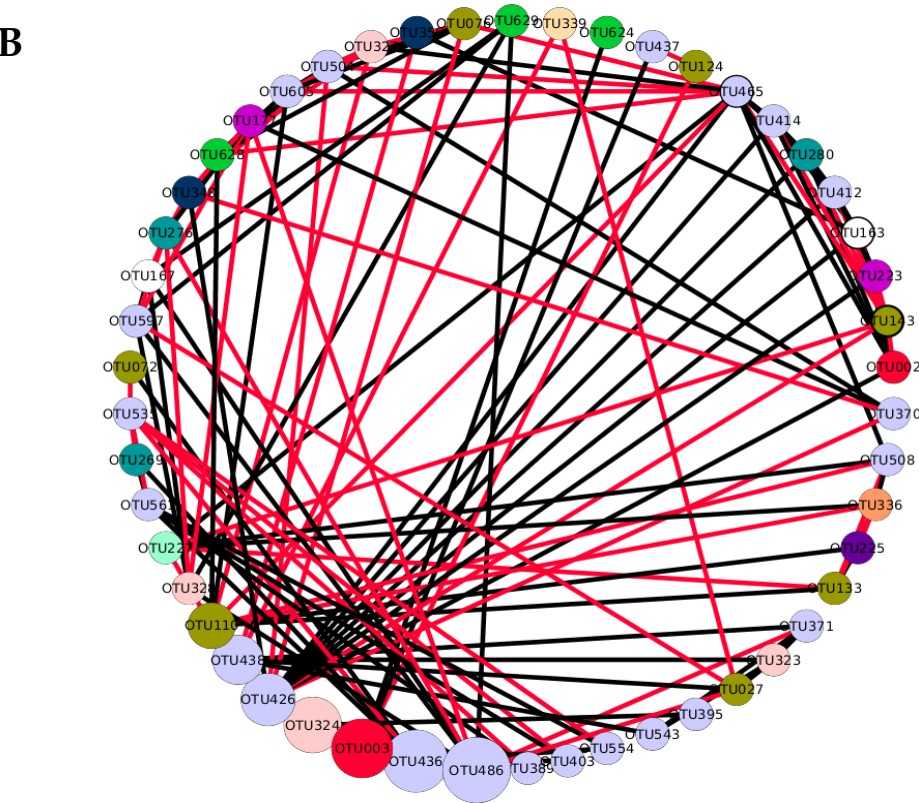

**C**

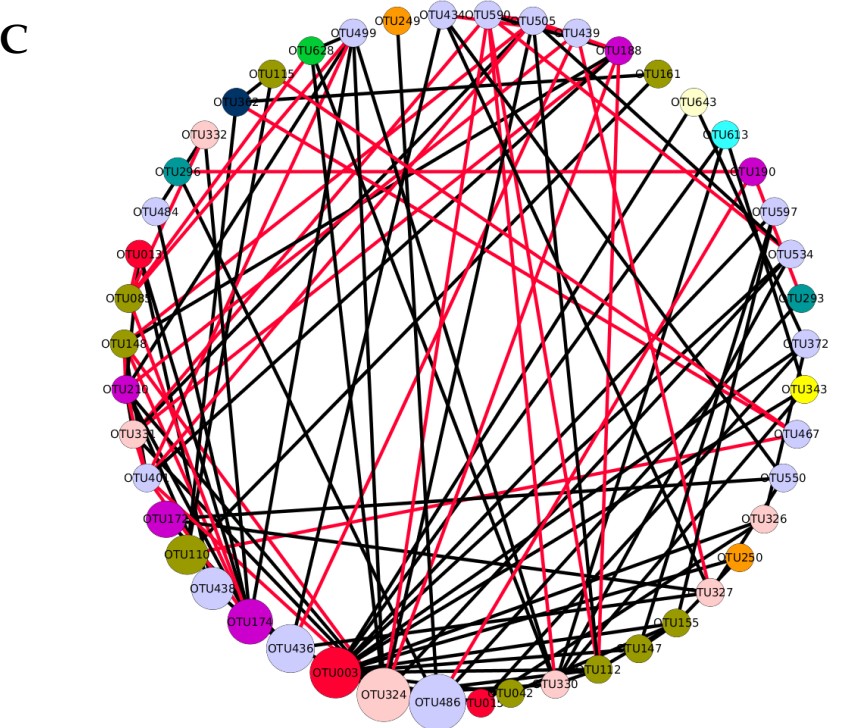

**Figure 2.** Subnetwork of the eight most abundant OTUs and their first neighbor nodes in the (**A**) conventional cropping system (Conv) and both organic cropping systems, (**B**) Org_C and (**C**) Org_M. Each circle represents an OTU, and its color represents a phylum. The size of the circle corresponds to the OTU's abundance; OTUs are arranged so that they neighbor OTUs of similar size. Red and black lines represent negative and positive edges.

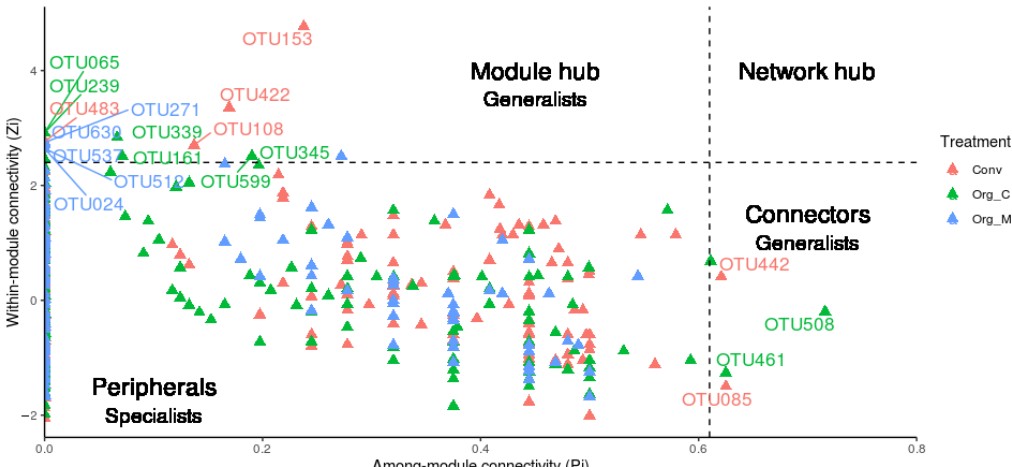

**Figure 3.** Zi-Pi plot showing the distribution of OTUs according to their topological roles. Each color represents the different cropping system nodes from the networks: Conv, conventional cropping system; Org_C, organic cropping system with compost and compost tea; Org_M, organic cropping system with manure; OTU065, OTU524, OTU161, OTU153, OTU108, OTU085, Actinobacteria; OTU239, Chloroflexi; OTU483, OTU537, OTU512, OTU422, OTU599, OTU442, OTU508, OTU461, Proteobacteria; OTU630, Verrucomicrobia; OTU271, Firmicutes; OTU339, NKB19, OTU345, Planctomycetes.

### 3.3. Predictive Functional Community

Closed-reference OTU picking resulted in 5.497 OTUs, which were classified into 6.909 predictive functional categories (Table S5). The majority ($\approx$60%) of the functional genes were assigned to metabolism, followed by genetic information processing ($\approx$19%), environmental information processing ($\approx$15%) and cellular processes and organismal systems (0–5%). In general, pathways related to the following functional categories were higher in Org_C than in Conv and Org_M: carbon fixation, nitrogen metabolism, amino acid metabolism/enzymes, lipid metabolism, bacterial toxins, biosynthesis and the biodegradation of secondary metabolites, phosphotransferase system PTS, the signal transduction mechanism, tetracycline biosynthesis, toluene degradation, sulfur metabolism, phenylalanine, tyrosine and tryptophan biosynthesis (Table S5). In Org_C, benzoate degradation, DNA-replication proteins and carbon-fixation pathways in prokaryotes showed the highest values (Table S5).

### 3.4. Predictive Nitrogen Functional Community

Analysis of the most abundant N metabolism pathway genes revealed that the relative abundance of $N_2$-fixing functional genes (*nifH*, *nifD* and *nifK*) was significantly higher in Org_C than in Org_M, while no significant difference was observed with Conv, with the exception of the *nifK* gene, which showed higher values in Org_C (Table 2). Predicted denitrification genes, such as the denitrifying nitrous oxide reductase gene (*nosZ*) or nitrate reductase (*narG*), did not show any significant differences between the three cropping systems, although Org_C did have higher values (Table 2). Nitric oxide reductase (*norB*) and nitrite reductase (*nirK*) showed significantly higher predicted abundance in Org_C and Conv than in Org_M, while ammonium-forming nitrite reductase (*nrfA*) and nitrate reductase (*narH*) showed higher values in Org_C than in Org_M (Table 2). In the nitrification process, the ammonia oxidation-predicted genes (*amoA*/*amoB* and *amoC*) were significantly increased in Conv and Org_C compared to Org_M, while the nitrification functional-predicted gene hydroxylamine oxidoreductase (*Hao*) was increased significantly in Org_C compared to Org_M and Conv (Table 2).

**Table 2.** Predicted N gene count for the most abundant N cycling genes detected in three cropping systems.

| | Gene | Conv | Org_C | Org_M | ANOVA | Kruskal—Wallis |
|---|---|---|---|---|---|---|
| N-fixation | *nifH* | 0.42 ± 0.06 ab | 0.47 ± 0.07 a | 0.32 ± 0.10 b | * | - |
| | *nifD* | 0.43 ± 0.07 ab | 0.52 ± 0.06 a | 0.36 ± 0.06 b | ** | - |
| | *nifK* | 0.43 ± 0.03 b | 0.59 ± 0.13 a * | 0.28 ± 0.05 c * | *** | - |
| Denitrification | *narG* | 0.38 ± 0.06 | 0.49 ± 0.11 | 0.36 ± 0.14 | ns | - |
| | *narH* | 0.38 ± 0.05 b | 0.49 ± 0.05 a | 0.36 ± 0.04 b | - | ** |
| | *nrfA* | 0.09 ± 0.00 ab | 0.15 ± 0.07 a | 0.07 ± 0.02 b | * | - |
| | *nirK* | 0.35 ± 0.08 a | 0.32 ± 0.09 a | 0.19 ± 0.05 b * | * | - |
| | *norB* | 0.48 ± 0.06 a | 0.43 ± 0.05 a | 0.29 ± 0.04 b *** | *** | - |
| | *nosZ* | 0.20 ± 0.03 | 0.25 ± 0.07 | 0.16 ± 0.04 | ns | - |
| Nitrification | *amoC* | 0.01 ± 0.00 a | 0.01 ± 0.00 a | 0.00 ± 0.00 b | - | *** |
| | *amoB* | 0.01 ± 0.00 a | 0.01 ± 0.00 a | 0.00 ± 0.00 b | - | *** |
| | *amoA* | 0.01 ± 0.00 a | 0.01 ± 0.00 a | 0.00 ± 0.00 b | - | *** |
| | *hao* | 0.08 ± 0.01 b | 0.13 ± 0.01 a | 0.08 ± 0.01 b | - | ** |

Values (mean ± sd; n = 5) are expressed with an e-value of $1 \times 10^{-2}$. In each cropping system, the mean value followed by *, ** or *** represents significant differences with respect to the conventional cropping system by Dunnett's test (* $p < 0.05$; ** $p < 0.01$; *** $p < 0.001$); missing asterisks denote non-significant differences. Different letters represent significant differences between cropping systems by Tukey's test or Dunn's Kruskal—Wallis multiple comparison test; Conv, conventional system; Org_C, organic cultivation with sheep manure compost and compost tea; Org_M, organic cultivation with sheep manure.

### 3.5. Soil Properties

Several chemical properties of the soils studied are shown in Table 3. Org_C soils showed the highest values of TN, NH$_4^+$ and TOC as well as some minerals, such as available Mg, K, Na and Ca. Org_M, on the other hand, showed the highest pH value. Available P was the highest in Conv, followed by Org_C and Org_M.

**Table 3.** Soil properties in the three cropping systems.

| Soil Properties | Cropping System | | | | |
|---|---|---|---|---|---|
| | Conv | Org_C | Org_M | ANOVA | Kruskal—Wallis |
| pH | 8.39 ± 0.17 b | 8.47 ± 0.14 ab | 8.70 ± 0.10 a | * | - |
| EC (dS m$^{-1}$) | 0.54 ± 0.15 | 0.52 ± 0.13 | 0.38 ± 0.04 | - | ns |
| TOC (g kg$^{-1}$) | 11.49 ± 0.28 ab | 15.64 ± 3.37 a | 9.01 ± 3.49 b | ** | - |
| TN (g kg$^{-1}$) | 1.13 ± 0.19 b | 1.59 ± 0.34 a | 0.93 ± 0.24 b | ** | - |
| *p* (mg kg$^{-1}$) | 20.15 ± 5.24 | 14.65 ± 7.71 | 14.33 ± 7.48 | ns | - |
| NH$_4^+$ (mg kg$^{-1}$) | 0.10 ± 0.23 b | 1.33 ± 0.15 b | 0.00 ± 0.00 a | - | ** |
| Mg (cmol kg$^{-1}$) | 3.54 ± 0.11 ab | 4.39 ± 1.09 a | 3.13 ± 0.54 b | * | - |
| K (cmol kg$^{-1}$) | 0.62 ± 0.15 | 0.85 ± 0.17 | 0.78 ± 0.06 | * | - |
| Na (cmol kg$^{-1}$) | 2.12 ± 0.32 | 2.19 ± 0.86 | 1.64 ± 0.23 | ns | - |
| Ca (cmol kg$^{-1}$) | 8.44 ± 0.83 | 10.03 ± 2.40 | 7.19 ± 1.49 | - | ns |

Values (mean ± sd; n = 5) followed by different lowercase letters correspond to significant differences between cultivation systems (Tukey's test or pairwise *t*-test by groups); (ns) non-significant differences between cultivation systems. The (-) symbol indicates the test did not proceed; significant levels: ** $p < 0.01$; * $p < 0.05$. Conv, conventional system; Org_C, organic cultivation with sheep manure compost and compost tea; Org_M, organic cultivation with sheep manure. TN, total nitrogen; TOC, total organic carbon; Mg, K, P, Na and Ca; available Mg, K, P, Na and Ca, respectively; EC, electrical conductivity.

### 3.6. Module and Node Correlations with Soil Properties

The heat-map representation of correlations among modules from pMENs and soil variables (Figure 4) showed that Org_C had the highest significant correlation ($p < 0.05$) among modules and soil properties, with 17 significant correlations (11 positive), followed by Org_M, with 11 significant correlations (2 positive), and Conv, with 5 significant correlations (3 positive) (Figure 4). TOC and TN only showed significant correlations within

organic cropping systems (Org_C and Org_M) (Figure 4). A Mantel test showed that soil properties have significant effects on the microbial communities in each cropping system, whereas they only affect functionality in the Conv cropping system (Table S6).

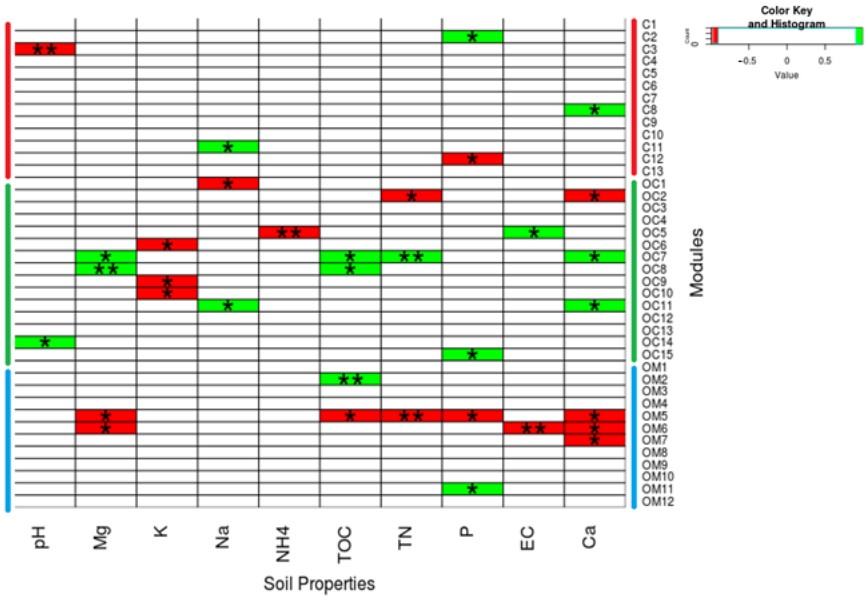

**Figure 4.** Heat map showing significant module correlation between soil properties and the modules in the Org_C, Org_M and Conv cropping systems. Colors represent positive (green) or negative (red) correlations, and the star symbol (*) in the cells represents the significance of that correlation, with *, $p < 0.05$; **, $p < 0.01$. Conv, conventional cropping system; Org_C, organic cropping system with compost and compost tea; Org_M, organic cropping system with manure; TN, total nitrogen; TOC, total organic carbon; Mg, K, P, Na and Ca; available Mg, K, P, Na and Ca, respectively; EC, electrical conductivity.

## 4. Discussion

Despite the importance of organic farming in sustainable agriculture, little research has explored the co-occurrence patterns among microbial groups or their functions. In this study, two long-term organic farming systems and a conventional system were studied to explore the co-occurrence patterns among soil bacterial taxa, N cycle functions and their relationship with chemical soil properties, with the aim of deepening our understanding of these microbial communities. Previously, Cuartero et al. [29] studied the microbial species, abundance and diversity of these agricultural systems, observing that the bacterial community structure changed according to the cropping system, although no significant differences were observed in diversity indices.

A microbial network study mainly analyzes the interactions among different microbial species in the soil to maintain ecosystem stability [12], considering that microorganisms are distributed into trophic levels or niches based on their nutritional preferences and functionality [39,40]. Network parameters indicate high modularity values, which means that the microbial community shows modular behavior [41] and can fully adapt to environmental changes [42]. These parameters also indicate habitat heterogeneity, non-random interaction patterns and ecological network complexity [33]. Overall, this approach allowed us to conclude that soil microorganisms tended to co-occur more than would be expected by chance [24]. The bacterial community in the Org_C network is potentially more complex and interconnected than the other two network systems (Org_M and Conv), probably due to the incorporation of compost, which is stable organic matter [43–46]. This suggests higher microbial cooperation and a greater exchange of metabolites and information among microbial species, probably due to the creation of favorable and stable niches that

select specific microbial taxa and build up more intensive interactions within the microbial community, which makes those networks more efficient when faced with disturbances. Moreover, a stable, nutrient-rich soil, such as Org_C [29], may contribute to more effective plant growth, nutrient cycling, C utilization, pathogen suppression [47] and the promotion or generation of more functional traits, rather than functional diversification [48]. On the other hand, the decrease in modularity in Conv and Org_M suggests inhibition of microbial functional diversity [49], which could be due to the higher amount of total pesticides in Conv, as previously observed by [50], and the higher pH in Org_M due to the incorporation of fresh organic matter [10,51].

Global networking showed that the cropping system with organic matter had more negative interactions among nodes. Negative interactions could suggest competition, exclusion or even preference for different niches [52,53]. However, according to Coyte et al. [54], negative interactions lead to network stabilization since they compensate for the overexpression of some members, which could lead to destabilization of the network. On the other hand, the eight most abundant OTUs, which showed inverse connections within the global network, also had a greater number of positive connections than within the conventional cropping system. A greater number of positive connections implies cooperativity or codependence among the nodes [54]; more concretely, positive correlations among species mean that these species' coexistence is based on more than chance (e.g., mutualism, predation, etc.). Therefore, our network analysis revealed that an organic-system microorganism habitat with more specialized niches gives stability to the microbial community and resistance to external factors, with cooperation amongst the most-abundant OTUs promoting the development of the other microorganisms. However, the Conv system showed a higher number of positive connections on its global network, which could indicate a lack of negative regulators, thus being prone to imbalance through external factors.

Modules of highly interconnected nodes [12,55] showed correlations with soil parameters. Org_C showed the highest correlations, indicating the influence of different soil parameters in this network topology, interactions and the potential ecosystem-level functions of the soil. Moreover, the high number of correlations could be due to a greater number of niches in Org_C, which could allow a greater number of different habitats. However, this is not entirely in accordance with the increased network complexity of Org_C, which suggests that module members are less affected by environmental perturbations [56,57]. Total nitrogen correlated positively with a module from Org_C (OC7), and this module was composed of Nitrospirae, where members of *Nitrospira* show the capability to perform complete nitrification [58,59], or Actinobacteria, where genera like *Streptomyces* have been linked to some nitrogen-fixation genes [60,61]. The TOC and TN content can be considered drivers of changes in the microbial community in Org_C and Org_M due to the incorporation of organic matter [62,63]. Quilty et al. [64] and Zhou et al. [57] suggested that the amendments had a selective effect on the bacterial community and were more obvious in bacteria than fungi because bacteria are more sensitive to the cropping system and the increase in organic carbon [65].

The identification of keystone bacterial populations is a critical issue in ecology, but is very difficult to achieve due to extreme complexity, high diversity and an uncultivated status [12]. Generalists are the key microorganisms in the microbial network and play important roles in it as predicted from network theory [23]. Generalist habitats have much higher environmental tolerances than specialist habitats, which are more restrictive [66], although generalists normally occupy a small fraction of modules [13,45], as was observed in our study. In our study, the Org_C network had the highest number of generalists, and it would therefore seem logical that this system would show better node communication within and/or among modules than the other two systems. Most generalists were composed of a diverse range of phylogenetic groups typical in soils, such as Proteobacteria, Actinobacteria, Planctomycetes, Chloroflexi, Firmicutes and Verrucomicrobia [67]. However, compared to the conventional cropping system, the incorporation of organic matter into the soil promoted some specific generalists, including Firmicutes (*Bacillus*) and

Verrucomicrobia in Org_M, and Planctomycetes, Chloroflexi and NKB19 in Org_C. Some of these generalists are plant-beneficial microorganisms, such as *Bacillus*, which has been described as a soil-borne pathogen inhibitor [68,69] through the production of antifungal compounds [70], or Planctomycetes and Chloroflexi, which can participate in complex organic matter degradation [71,72], increasing nutrient uptake in plants.

Furthermore, the roles of some nodes shifted in the three networks systems: nodes belonging to Proteobacteria and Actinobacteria, which were considered generalist in the Conv network, were considered specialist in the Org_C and Org_M networks. Generalist and specialist habitats show non-random co-occurrence patterns; furthermore, specialists have a greater and more robust structure than generalists, and these changes could be driven by deterministic processes [73]. This suggests that organic amendments may change the ecological roles of key bacteria [74] due to their wide capacity for adaptation and functionality [75,76].

Our results suggest that changes in bacterial habitats induced by organic matter could lead to different predicted functional groups depending on the new niches generated [77]. Org_C exhibited a higher abundance of different functional groups, which could indicate a more sustainable soil microbial community structure and higher microbial functionality [10]. In addition, Org_C showed a large increase in carbon fixation in the prokaryotes pathway, which hosts many kinds of autotrophic bacteria [78], including bacteria with $CO_2$-fixation capacity. This increase suggests that compost addition could contribute to $CO_2$ sequestration and storage as described by Ryals et al. [79]. Besides that, the nitrogen flow is considered very efficient and effective when microorganisms are actively transforming the organic nitrogen at the same time that plants are rapidly taking up the $NH_4^+$ and $NO_3^-$, as the potential for nitrogen loss is relatively low. In our experiment, N cycling was affected by the different cropping systems, and the abundance of gene families involved could predict the activity of N in the organic and conventional cropping systems, which is crucial, considering nitrogen is a key driver of soil microbial community composition [80]. The higher amount of *nifH*, the key marker gene for nitrogen fixation, in Org_C than in the other systems helps determine soil fertility [81] and could suggest the role of the compost. Moreover, fresh organic matter and inorganic fertilizer would inhibit the predicted N-fixation genes of specific N-fixer groups [82,83]. Genes involved in denitrification were more abundant than nitrification genes. However, genes involved in the nitrification process—principally hydroxylamine oxidoreductase (*hao*) involved in conversion of ($NH_2OH$) in nitrous oxide ($N_2O$)—were more abundant in Org_C. Denitrification is the basic avenue for nitrogen loss in agricultural soils [84,85], and the genes *narG*, *nirK*, *norB*, *nosZ* and *nrf* are involved in the conversion of nitrite to nitrogen gas. Org_C and Org_M showed lower values than Conv, although *nosZ*, the enzyme known to catalyze the last step of denitrification, the conversion of nitrous oxide ($N_2O$) to nitrogen gas ($N_2$), was higher in Org_C. This could indicate a decrease in $N_2O$ emissions [86]. Moreover, Org_C showed a higher predicted abundance of gene *nrfA*, related to nitrate reduction to ammonium (DNRA), which is beneficial to N retention and immobilization in agricultural soils since N is converted to $NH_4^+$ rather than lost through denitrification and anammox [87].

## 5. Conclusions

In this work, we provide insight into a soil bacterial community affected by different long-term cropping systems via network interaction analysis and functional analysis, principally N cycling. The network revealed how long-term compost application modified the bacterial community, increasing the network complexity and enhancing modulation and communication through generalists to a greater extent than in the Conv and Org_M cropping systems. In addition, changes in these bacterial habitats could have also altered bacterial functions, since Org_C showed higher predicted nitrogen-fixing potential, decreased $N_2O$ emissions and greater carbon-sequestration potential than the other cropping systems. This implies the importance of using a stable organic amendment as compost and

the use of adequate tools to study the response of the bacterial community to long-term cropping systems.

**Supplementary Materials:** The following supporting information can be downloaded at: https://www.mdpi.com/article/10.3390/agronomy12020316/s1, Table S1: Management characteristics of the three cropping systems; Table S2: Abundance of phyla in the different modules (expressed in percent); Table S3: Subnetworks of the eight most abundant OTUs and their first neighbor nodes in the three cropping systems; Table S4: Classification of generalists in the three cropping systems; Table S5: Predicted functions of the bacterial communities found in the three cropping systems (relative abundances); Table S6: Mantel analysis of relationships between bacterial community and soil properties and N cycling genes.

**Author Contributions:** Conceptualization, J.C., V.S.-N., R.Z., J.W., J.A.P., J.-M.V. and M.R.; methodology, J.C., O.Ö., V.S.-N., J.W., R.Z., J.A.P., J.-M.V. and M.R.; validation, R.Z., J.A.P. and M.R.; formal analysis, J.C. and J.-M.V.; investigation, J.C., O.Ö., V.S.-N., J.W., R.Z., J.A.P., J.-M.V. and M.R.; resources, J.C., O.Ö., V.S.-N., J.W., R.Z., J.A.P., J.-M.V. and M.R.; data curation, J.C. and J.-M.V.; writing—original draft preparation, J.C.; writing—review and editing, J.A.P., J.-M.V. and M.R.; visualization, J.C. and J.-M.V.; supervision, J.A.P., J.-M.V. and M.R.; project administration, J.A.P. and M.R.; funding acquisition, J.A.P. and M.R. All authors have read and agreed to the published version of the manuscript.

**Funding:** The project was financially supported by the European Union (H2020 program–728003-Diverfarming).

**Institutional Review Board Statement:** Not applicable.

**Informed Consent Statement:** Not applicable.

**Data Availability Statement:** Data available in Zenodo: 10.5281/zenodo.4763091.

**Acknowledgments:** Thanks to Ansley Evans for her English editing.

**Conflicts of Interest:** The authors declare no conflict of interest.

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
