# Peer review of "Long-Term Compost Amendment Changes Interactions and Specialization in the Soil Bacterial Community, Increasing the Presence of Beneficial N-Cycling Genes in the Soil"

_agronomy, doi:10.3390/agronomy12020316_

Round 1

Reviewer 1 Report

ll. 1-4 Please strike or reformulate the part 'activating N-cycling genes' - this is not shown in the presented work and only based on function prediction of OTUs

ll. 25-27 predicted functionality, higher referred functional numbers of nitrogen fixation related genes

ll. 99-114 Did you compare replicates? Soil sample replicates can vary greatly. Please comment on this, if you did replicates and if not, why.

l. 131  The MENA pipeline is not accessible for me under the given link http://ieg2.ou.edu/MENA/, the server at ieg2.ou.edu is taking too long to respond. Is there maybe a github link or else you can provide in your manuscript?

ll. 279 -281 The authors speak of phyla, was the network built on phylum or OTU (mixed) level?

ll. 352-356 Figure 2 would profit a lot in readability and relevance if you provide a legend for the OTU phylogenetic assignments side by side or label the OTU nodes with their lowest rank phylogenetic assignment

ll 376-379 This Figure 3 is very informative! Same as for Figure 2, it will be even more informative and easy accessible if you provide phylogenetic assignments of the relevant OTU right in the figure.

l. 396 followed by

l. 409 Predicted denitrification genes

l. 412 predicted abundance

ll 414-417 The predicted gene abundance increases, not the gene itself

l. 521 being exposed

ll. 529-531 Some taxa denotations miss the cursive/italics

ll. 545-553 Taxa denotations miss the cursive/italics

l. 555 Taxa denotations miss the cursive/italics

ll. 562-563 predicted functional groups

l. 573 predicted amount

l. 576,577 Predicted genes

l 584 higher predicted abundance of genE

ll. 561-586 Please mind the difference between genes and gene products and that gene names are generally written in italics

I am missing a paragraph denoted to the 16S rRNA gene sequencing bias and the the bias and limitations of the function prediction referred from OTUs

l.595 higher predicted/referred nitrogen-fixing potential

Author Response

Point 1. ll. 1-4 Please strike or reformulate the part 'activating N-cycling genes' - this is not shown in the presented work and only based on function prediction of OTUs.

Answer 1. Thank you for your suggestion, I have changed “activating N-cycling genes” to “increasing the presence of”.

Point 2. 25-27 predicted functionality, higher referred functional numbers of nitrogen fixation related genes.

Answer 2. Thank you for your suggestion, I have added it to the manuscript.

Point 3. 99-114 Did you compare replicates? Soil sample replicates can vary greatly. Please comment on this, if you did replicates and if not, why.

Answer 3. We compare treatments and each treatment has replicates. Afterwards we make Statistical analysis with all the replicates from the treatments as the statistic need it.

Point 4. 131 The MENA pipeline is not accessible for me under the given link http://ieg2.ou.edu/MENA/, the server at ieg2.ou.edu is taking too long to respond. Is there may be a github link or else you can provide in your manuscript?

Answer 4. Sorry for this issue, I have changed the link so this should work correctly: http://ieg4.rccc.ou.edu/mena/. Thank you for your supervision.

Point 5. 279 -281 The authors speak of phyla, was the network built on phylum or OTU (mixed) level?

Answer 5. The network was elaborated with OTUS identified at Phylum level.

Point 6. 352-356 Figure 2 would profit a lot in readability and relevance if you provide a legend for the OTU phylogenetic assignments side by side or label the OTU nodes with their lowest rank phylogenetic assignment

Answer 6. Thank you for your suggestion, I have added a common legend with the taxonomy at phyla level.

Point 7.  376-379 This Figure 3 is very informative! Same as for Figure 2, it will be even more informative and easy accessible if you provide phylogenetic assignments of the relevant OTU right in the figure.

Answer 7. Thank you for your suggestion, I have added taxonomy classification in the footnote of the figure to facilitate the interpretation.

Point 8. 396 followed by

Answer 8. It has been changed as you suggested, Thank you

Point 9. 409 Predicted denitrification genes

Answer 9. It has been changed as you suggested, Thank you

Point 10. 412 predicted abundance

Answer 10. It has been changed as you suggested, Thank you

Point 11414-417 The predicted gene abundance increases, not the gene itself

Answer 11. It has been changed as you suggested, Thank you

Point 12521 being

Answer 12. It has been changed as you suggested, Thank you (Lin 514)

Point 13. 529-531 Some taxa denotations miss the cursive/italics

Answer 13. Many thanks for your observation, Nitrospirae and Antinobacteria are both considered as phyla, so we have not italized them to differentiate from genus and species classification.

Point 14. 562-563 predicted functional groups

Answer 14. It has been changed as you suggested, Thank you

Point 15. 573 predicted amount

Answer 15. It has been changed as you suggested, Thank you

Point 16. 576,577 Predicted genes

Answer 16. It has been changed as you suggested, Thank you

Point 17. 584 higher predicted abundance of genE

Answer 17. It has been changed as you suggested, Thank you

Point 18. 561-586 Please mind the difference between genes and gene products and that gene names are generally written in italics

Answer 18. It has been carefully changed along the manuscript as you suggested, Thank you.

Point 19. I am missing a paragraph denoted to the 16S rRNA gene sequencing bias and the the bias and limitations of the function prediction referred from OTUs

Answer 19. Thank you for your suggestion, I have added this sentence to the manuscript “but the limitations of metagenomic technique must be taking into account, mainly due to short length of the fragment, as well as prediction accuracy so we should consider it as a potential functionality.”

Point 20. 595 higher predicted/referred nitrogen-fixing pot

Answer 20. It has been changed as you suggested, Thank you

Thank you very much for your revision and your words, we considerer that the quality of our manuscript has been improved.

Reviewer 2 Report

It is generally recognized the importance of microorganisms for soil fertility is extremely high. This issue is essential for agroecosystems, which are artificial and require human assistance to maintain their stability and productivity. There are different approaches to the study of soil microbial communities (microbiomes) and microbial interactions. As a rule, they are limited to the assessment of species diversity in different soils caused by various factors. Such approaches do not always provide an opportunity to assess the diverse relationships between microorganisms in communities and their functions in ecosystems.

In the previous study (Cuartero et al., 2021), the authors identified differences in the phylogeny of soil microbial communities when using various farming systems. In this work, the authors have chosen cutting-edge and adequate methods. Firstly, the methods of molecular biology and metagenome analysis were used to obtain data on the taxonomy of microorganisms, and secondly, the methods of phylogenetic molecular ecological networks made it possible to identify and confirm some patterns. The reliability of the results is confirmed by high-quality statistical analysis.

The authors convincingly demonstrated the changes in the microbiomes by the long-time action of organic additives, due to the complication of the microbial community and its stabilization. I would like to congratulate the authors. They have done a great job.

Author Response

It is generally recognized the importance of microorganisms for soil fertility is extremely high. This issue is essential for agroecosystems, which are artificial and require human assistance to maintain their stability and productivity. There are different approaches to the study of soil microbial communities (microbiomes) and microbial interactions. As a rule, they are limited to the assessment of species diversity in different soils caused by various factors. Such approaches do not always provide an opportunity to assess the diverse relationships between microorganisms in communities and their functions in ecosystems.

In the previous study (Cuartero et al., 2021), the authors identified differences in the phylogeny of soil microbial communities when using various farming systems. In this work, the authors have chosen cutting-edge and adequate methods. Firstly, the methods of molecular biology and metagenome analysis were used to obtain data on the taxonomy of microorganisms, and secondly, the methods of phylogenetic molecular ecological networks made it possible to identify and confirm some patterns. The reliability of the results is confirmed by high-quality statistical analysis.

The authors convincingly demonstrated the changes in the microbiomes by the long-time action of organic additives, due to the complication of the microbial community and its stabilization. I would like to congratulate the authors. They have done a great job.

Thank you very much for your revision and your words. We are very grateful for the time you spent to review our manuscript and very happy that you liked it, have a good day.